# *Circaea mollis* Siebold & Zucc. Induces Apoptosis in Colorectal Cancer Cells by Inhibiting c-Myc Through the Mediation of RPL5

**DOI:** 10.3390/ijms26104664

**Published:** 2025-05-13

**Authors:** So-Mi Park, Nanyeong Kim, Ye-Rin Park, Seok Woo Kim, Ji Hoon Jung, Yun-Cheol Na, Daeho Kwon, Hyungsuk Kim, Hyeung-Jin Jang

**Affiliations:** 1College of Korean Medicine, Kyung Hee University, Seoul 02447, Republic of Korea; psm991030@naver.com (S.-M.P.); knylike@naver.com (N.K.); dutndi@naver.com (Y.-R.P.); kim66470@naver.com (S.W.K.); johnsperfume@gmail.com (J.H.J.); 2Department of Science in Korean Medicine, Graduate School, Kyung Hee University, Seoul 02447, Republic of Korea; 3Metropolitan Seoul Center, Korea Basic Science Institute, Seoul 03759, Republic of Korea; nyc@kbsi.re.kr; 4Department of Microbiology, College of Medicine, Catholic Kwandong University, Gangneung 25601, Republic of Korea; dkwon@cku.ac.kr; 5Department of Korean Rehabilitation Medicine, Kyung Hee University Medical Center, Seoul 02447, Republic of Korea; kim0874@hanmail.net

**Keywords:** *Circaea mollis* Siebold & Zucc., colorectal cancer, apoptosis, c-Myc, oncogene, RPL5

## Abstract

Colorectal cancer remains a significant global health concern. In this study, we investigated the anticancer potential of *Circaea mollis* Siebold & Zucc. (CS&Z), a traditional medicinal plant known for its anti-inflammatory, anti-arthritic, and antioxidant properties, in the treatment of colorectal cancer. We found that CS&Z induces apoptosis and G1/S phase cell cycle arrest in colorectal cancer cells, primarily through the suppression of the proto-oncogene c-Myc. Specifically, the depletion of RPL5, a ribosomal protein associated with c-Myc regulation, reversed the suppression of c-Myc by CS&Z. Additionally, when co-administered with the standard chemotherapeutic agents doxorubicin and 5-fluorouracil, CS&Z demonstrated synergistic effects, thereby further emphasizing its potential efficacy as a therapeutic option for the treatment of colorectal cancer. Moreover, the constituents of CS&Z, detected through liquid chromatography–mass spectrometry analysis, reportedly exhibit anticancer activities. Taken together, our findings suggest that CS&Z holds promise as a natural product capable of modulating oncogenic signaling in colorectal cancer and may serve as a complementary agent in future therapeutic strategies.

## 1. Introduction

Colorectal cancer currently accounts for approximately 10% of cancer-related deaths [1] and, according to the GLOBOCAN 2020 report, it is the third most common type of cancer worldwide in terms of new cases [2]. Approximately 54.2% of colorectal cancer deaths occur in Asia, and in South Korea, among cancers, it has the second-highest incidence following thyroid cancer [3]. Although colorectal cancer (CRC) is generally more common in older adults, recent epidemiological data show a 30-year decline in incidence among individuals aged 50 and older, accompanied by a concerning rise in cases among adults under 50 years old [4,5]. The development of colorectal cancer has been attributed to a diverse range of factors, including genetics, obesity, smoking, and alcohol consumption [6], and current treatment options include chemotherapy, radiation therapy, surgery, and targeted therapy [7]. Among these, chemotherapy is the primary treatment approach, particularly using drugs such as doxorubicin (DOX), oxaliplatin, and 5-fluorouracil (5-FU) [8,9,10], although more recently, there is also a growing trend toward therapies that regulate identified oncogenes.

Among such targeted genes commonly expressed, not only in colorectal cancer but also in other types of cancer, such as pancreatic and breast cancers [11], is c-Myc, a proto-oncogene that plays a key role in the initiation and progression of most cancers [12]. c-Myc is involved in processes essential to cancer development, including transcription, growth, proliferation, cell cycle regulation, metabolism, and cell apoptosis [13]. Given its critical role in regulating cell proliferation, apoptosis, and ribosome biogenesis, and its frequent overexpression in colorectal cancer, *c-Myc* was selected as a key oncogenic target in this study. In particular, its known regulatory relationship with *RPL5* made it a mechanistically relevant focus for evaluating the anticancer effects of *Circaea mollis* Siebold & Zucc. (CS&Z) [14,15].

Further research has focused on cancer-associated apoptosis by examining the levels of key proteins implicated in the apoptotic process, including poly (adenosine diphosphate-ribose) polymerase (PARP) and caspase 3. In addition, Bcl-extra-large (Bcl-xL), a member of the Bcl-2 protein family, is involved in the regulation of the intrinsic apoptotic pathway [16]. Bcl-2 proteins have been associated with apoptosis by binding to and inactivating Bax and Bak, thereby contributing to regulation of the caspase cascade [16]. Furthermore, among the oncogenes identified as being associated with the regulation of c-Myc, CNOT2, a subunit of the CCR4-NOT complex (CNOT), has been established to play roles in apoptosis, angiogenesis, and proliferation in a range of cancer types [10]. MID1IP1, a further oncogene implicated in the regulation of c-Myc, has been identified as a negative regulator of AMP-activated protein kinase (AMPK), and recent research has revealed that a depletion of MID1IP1 leads to the inhibition of c-Myc in colorectal and liver cancer growth [14], and that the inhibition of CNOT2 induces apoptosis via MID1IP1 [17]. A further characteristic of apoptosis is the induction of cell cycle arrest, and accordingly, a number of studies have examined the expression of G1/S phase-related factors in this context, including cyclin E1, cyclin D1, and cyclin-dependent kinase 6 (CDK6) [18].

CS&Z, a flowering plant, is a member of the evening primrose family Onagraceae, certain species of which exhibit medicinal properties, including anti-inflammatory, anti-arthritic, and antioxidant activities [19]. It is primarily distributed in North America and Eurasia [20]. For example, among plants in the genus *Circaea*, *Circaea lutetiana* L. has antimicrobial and anti-inflammatory properties [19]. Moreover, in the case of CS&Z, recent research has demonstrated the alleviation of post-menopausal osteoporosis in a mouse model via the BMP-2/4/Runx2 pathway [21]. Additionally, in an inflammatory swelling model, preparations of this plant demonstrate anti-inflammatory activity by reducing serum TNF-α and IL-1β levels, increasing serum IL-10, and reducing inflammatory infiltration [22]. Furthermore, CS&Z is widely used as a traditional herbal medicine in Hani ethnopharmacy [21], with major active constituents, including luteolin, apigenin, *p*-coumaric acid, astragalin, and vitexin, being identified [19,22].

In this study, in which we sought to assess the therapeutic potential of CS&Z in the treatment of colorectal cancer, cell lines were treated with CS&Z, and we examined its effects on the expression of a range of apoptotic factors, oncogenes, and cell cycle-related factors, particularly that of c-Myc. Additionally, we confirmed that the ribosomal protein RPL5 is involved in mediating the CS&Z-induced suppression of c-Myc expression, and also established that CS&Z acts synergistically when co-administered with the chemotherapeutic agents DOX or 5-FU. On the basis of our findings, we propose that CS&Z has potential application as a therapeutic option for the treatment of colorectal cancer.

## 2. Results

### 2.1. CS&Z Inhibits the Viability and Proliferation of Colorectal Cancer Cells

To determine whether CS&Z reduces the viability of colorectal cancer cells, we performed an MTT assay. We accordingly observed a dose-dependent reduction in the viability of cells treated with an extract of CS&Z, thereby indicating the cytotoxic activity of this extract (Figure 1A). In addition, to assess whether CS&Z influences the proliferation of colorectal cancer cells, we performed a colony formation assay. Similarly, observations revealed a dose-dependent reduction in the proliferation of cells treated with CS&Z (Figure 1B).

### 2.2. CS&Z Induces Apoptosis and Regulates Oncogenes Such as c-Myc in Colorectal Cancer Cells

On the basis of western blotting analysis, we demonstrated that CS&Z induces both dose-dependent (Figure 2A) and time-dependent (Figure 2B) changes in the protein levels of apoptotic factors, including c-Myc, pro-PARP, cleaved-PARP, pro-caspase 3, and Bcl-xL. Furthermore, based on previous research, we established that the inhibition of CNOT2 induces apoptosis via reductions in MID1IP1 and c-Myc, and we thus investigated whether CS&Z regulates the expression of the oncogenes MID1IP1 and CNOT2 in colorectal cancer cells [17]. Additionally, using a TUNEL assay, we confirmed that CS&Z induces apoptosis in these cells, as evidenced by an increase in the quantity of TUNEL-positive cells (Figure 2C). The data indicated apoptosis of the treated cells, as evidenced by the detection of DNA fragments. The nuclei of cells were stained blue using DAPI, whereas TUNEL-positive cells were stained green. In conclusion, by examining the protein levels of apoptotic factors and oncogenes in colorectal cancer cells treated with CS&Z, we confirmed the occurrence of apoptotic changes. Furthermore, by verifying DNA fragments using a TUNEL assay, we provided evidence of apoptotic induction.

### 2.3. CS&Z Induces Cell Cycle Arrest in Colorectal Cancer Cells at the G1/S Stage

To investigate whether CS&Z contributes to regulating the cell cycle in colorectal cancer cells, we examined the expression of factors involved in the G1/S phase using western blotting. The results revealed dose-dependent (Figure 3A) and time-dependent (Figure 3B) reductions in the expression of a range of cell cycle proteins, including cyclin E1, cyclin D1, and CDK6, accordingly indicating that CS&Z can influence the progression of the cell cycle in colorectal cancer cells.

### 2.4. CS&Z Inhibits c-Myc Sensitivity About FBS in Colorectal Cancer Cells

To investigate whether CS&Z regulates the expression of c-Myc in the response of colorectal cancer cells to serum stimulation, we exposed these cells to 0.2% and 20% FBS. Having initially synchronized the cell cycle using 0.2% FBS, we treated the cells with CS&Z diluted in 20% FBS and subsequently harvested these at different time points (0, 6, 12 and 24 h). The results revealed that, compared with the control cells treated with DMSO, the cells treated with CS&Z were characterized by lower levels of c-Myc expression (Figure 4). On the basis of these observations, we thus confirmed that CS&Z inhibits the sensitivity of c-Myc to serum-induced stimulation in colorectal cancer cells.

### 2.5. CS&Z Reduces c-Myc Stability in Colorectal Cancer Cells

To assess the stability of c-Myc in colorectal cancer cells when treated with CS&Z, we used cycloheximide. The selected cell lines were initially treated with CS&Z for 24 h and subsequently exposed to cycloheximide for different time intervals (0, 30, 60, and 90 min). Compared with the levels in the DMSO-treated controls, we detected lower levels of c-Myc expression in the cells treated with CS&Z (Figure 5), thereby indicating that, in the presence of cycloheximide, CS&Z attenuates the stability of c-Myc in colorectal cancer cells, as evidenced by the reduction in the half-life of this oncogene.

### 2.6. CS&Z Modulates c-Myc by Mediating RPL5 in Colorectal Cancer Cells

The ribosomal protein RPL5 can suppress the expression of c-Myc [23]. Therefore, to confirm whether the CS&Z-induced inhibition of c-Myc is mediated via RPL5 in colorectal cancer cells, we transfected cells with RPL5 siRNA and assessed the effects of silencing RPL5 on c-Myc expression. The results indeed revealed that the CS&Z-induced inhibition of c-Myc was reversed following the loss of RPL5 (Figure 6), thereby tending to indicate that the CS&Z-regulated expression of c-Myc in colorectal cancer cells is mediated via RPL5.

### 2.7. CS&Z Has a Synergic Antitumor Effect on Colorectal Cancer Cells When Co-Administered with 5-FU or DOX

Standard chemotherapy for colorectal cancer using agents such as 5-FU and DOX exhibits certain undesirable side effects, including the development of resistance, thereby highlighting the necessity to identify drugs that, when used in combination with existing anticancer agents, can contribute to conferring synergistic effects. To this end, we co-administered colorectal cancer cells with CS&Z in combination with either 5-FU or DOX and examined changes in cell viability and the expression of apoptotic factors and oncogenes. Using the MTT assay, we detected a significant reduction in the viability of the co-treated colorectal cancer cells with CS&Z compared with the viability of cells treated with 5-FU (Figure 7A) or DOX (Figure 7B) alone. Furthermore, on the basis of western blot analysis, we confirmed a reduction in the expression of apoptotic factors, including PARP, and caspase 3, as well as oncogenes such as c-Myc, CNOT2, and MID1IP1, in the co-treated cells with CS&Z, compared with the expression detected in cells treated with 5-FU or DOX alone (Figure 7C,D). We thus demonstrated that co-administration with CS&Z may enhance the efficacy of currently employed anticancer agents, such as DOX and 5-FU, in colorectal cancer cells.

### 2.8. LC-MS Analysis of a Methanolic Extract of CS&Z

To identify the active constituents of *C. mollis* that may contribute to the observed anticancer effects of CS&Z in colorectal cancer, we conducted LC-MS analysis. Most of the components were identified in both ESI+ and ESI- modes, and qualitative analysis was performed using ESI+: [M + H]+ and ESI−: [M + H]− ions. The LC-MS chromatogram obtained for the methanolic extract of CS&Z was characterized by two peaks, and we identified a total of five active constituents, namely galuteolin, isoorientin, orientin, astragalin, and vitexin (Figure 8), which are summarized in Table 1.

## 3. Discussion

The therapeutic options currently available for the treatment of colorectal cancer are surgery, chemotherapy, radiation therapy, and targeted therapy, among which chemotherapy is considered essential, particularly subsequent to the second stage of colorectal cancer. However, although numerous studies have been conducted regarding the efficacies of the respective treatments, a definitive cure for colorectal cancer remains unestablished. Moreover, in addition to questions of efficacy, it has been widely reported that traditional chemotherapeutic agents commonly used in colorectal cancer treatment, such as doxorubicin and 5-fluorouracil, have notable drawbacks, including the development of resistance and associated side effects. Consequently, further research is required on alternative treatment strategies, including the use of herbal medicines with low toxicity. CS&Z has been identified as a candidate for such an approach, demonstrating for the first time its ability to induce apoptosis in colorectal cancer cells via *RPL5*-mediated inhibition of the oncogene c-Myc.

LC-MS analysis performed to identify the active constituents of *C*. *mollis* associated with its anticancer effects revealed five candidate active compounds, namely galuteolin, isoorientin, orientin, astragalin, and vitexin, the effects of which were subsequently investigated [22]. All five compounds are flavonoids, characterized by aromatic ring structures derived from phenylalanine and malonyl-coenzyme A. In plants, flavonoids contribute to the pigmentation of different organs, including flowers and fruits, and also play essential roles in signal transduction between plants and microorganisms, have antimicrobial properties, and are involved in ultraviolet protection. Additionally, studies have reported significant health-beneficial properties when consumed by animals, thereby generating considerable research interest in these compounds [24].

Numerous studies have reported that galuteolin, isoorientin, orientin, astragalin, and vitexin have anti-inflammatory, antioxidant, and anticancer properties, and they have notably been found to promote anticancer effects via different pathways in a wide range of cancers [22,25,26,27,28,29,30,31,32,33,34]. We accordingly suggested that CS&Z may also have anticancer effects. Although five distinct compounds were detected in peak 1 of Figure 8, their individual contributions to the observed anticancer effects remain unclear. This represents a limitation of the current study. Further fractionation and isolation studies are planned to identify and characterize the specific active constituents responsible for the therapeutic activity of CS&Z in colorectal cancer.

Assessment of the cytotoxicity of CS&Z revealed reductions in cell viability and proliferation. In this regard, the oncogene c-Myc, a member of the *MYC* transcription factor family, is abnormally expressed in up to 70% of human cancers [35]. Accordingly, identifying a means to inhibit c-Myc activity is an important focus area in cancer research. The findings of previous research in this respect have revealed that cleaved-PARP activates caspase 3 and thereby mediates the inhibition of c-Myc via ribosomal protein L5 (RPL5) [15], whereas the inhibition of CNOT2 and MID1IP1 has been shown to suppress the expression of c-Myc [14]. Building upon these findings, we targeted c-Myc in colorectal cancer cells using CS&Z, and accordingly detected a reduction in c-Myc protein levels, which was accompanied by apoptotic changes in the protein levels of PARP, caspase 3, CNOT2, and MID1IP1. Moreover, we also observed reductions in the expression of Bcl-xL, an anti-apoptotic factor in the Bcl-2 family of proteins [36].

Given the established role of cyclin-dependent kinases (CDKs) and cyclins in cell cycle regulation [37], we examined the expression of cyclin D1, cyclin E1, and CDK6 in colorectal cancer cells treated with CS&Z extract. Western blot analysis suggested that the extract may induce G1/S phase arrest. However, since these findings are based on indirect protein-level evidence, further studies using flow cytometry are necessary to accurately determine the specific point of cell cycle arrest.

It has previously been reported that the binding of RPL5 to c-Myc contributes to promoting ribosomal stress, which subsequently results in the inhibition of transcriptional activity [23]. Having initially confirmed a CS&Z-induced reduction in c-Myc expression in HCT116^p53+/+^ and DLD-1 cells, we observed that a depletion of RPL5 could reverse the reduction in c-Myc expression. Furthermore, in colorectal cancer cells, CS&Z demonstrated a decrease in sensitivity through the reduction of c-Myc induced by serum stimulation, and it also showed a decrease in stability when protein synthesis was inhibited by CHX. Notably, the cytotherapeutic agents DOX, which is associated with cardiotoxicity, and 5-FU, linked to gastrointestinal side-effects, are currently undergoing scrutiny from the perspective of cell resistance [15]. Therefore, we found that, when co-administered with either DOX or 5-FU in the treatment of colorectal cancer cells, CS&Z promoted a synergistic increase in apoptotic changes, thereby indicating the potential efficacy of such combination treatments as therapeutic options for colorectal cancer.

On the basis of our findings, we propose that CS&Z induces apoptosis in colorectal cancer cells, primarily via RPL5-mediated regulation of the oncogene c-Myc. Furthermore, CS&Z modulates the protein levels of apoptotic factors, oncogenes, and cell cycle-related factors in colorectal cancer cells, leading to apoptotic changes. Additionally, when combined with the currently used cytotherapeutic agents DOX and 5-FU, CS&Z may promote a synergistic effect in the treatment of colorectal cancer. Compared with other plant-derived anticancer agents, CS&Z targets the RPL5–c-Myc axis in colorectal cancer and also potentiates the effects of conventional chemotherapeutics, offering a potentially therapeutic advantage.

Although the present study was conducted in vitro, the findings may provide a foundation for future investigations into the biological activity and therapeutic applicability of CS&Z in vivo.

## 4. Materials and Methods

### 4.1. Reagents

Roswell Park Memorial Institute 1640 (RPMI-1640) medium was obtained from Corning, Inc. (New York, NY, USA); fetal bovine serum (FBS) was obtained from Gibco (Grand Island, CA, USA); penicillin–streptomycin was obtained from Thermo Fisher (Waltham, MA, USA); and phosphate-buffered saline was obtained from Corning, Inc. (New York, NY, USA). When conducting SDS-PAGE, we primarily used the Dyne Prestained Protein Ladder Marker (Seongnam, Republic of Korea). Primary antibodies were prepared by diluting to a concentration of 1:1000 in Tris-buffered saline (TBS) containing 0.1% Tween 20. Primary antibodies against MID1IP1 (Cat No. 15764-1) were purchased from ProteinTech Antibody Group (Chicago, IL, USA), and those against PARP (Cat No. 9542), CNOT2 (Cat No. 34214), Bcl-xL (Cat No. 2764), cyclin D1 (Cat No. 2922), and cyclin E1 (Cat No. 20808) were purchased from Cell Signaling Technology (Beverly, MA, USA). These were detected following binding to goat anti-rabbit IgG-HRP secondary antibodies (1:10,000). Primary antibodies against c-Myc (Cat No. 67447-1-Ig) were purchased from ProteinTech Antibody Group (Chicago, IL, USA); those against caspase 3 (Cat No. sc7272) and β-actin (Cat No. sc-47778) were purchased from Santa Cruz Biotechnology (Santa Cruz, CA, USA); and those against CDK6 (Cat No. 3136S) purchased from Cell Signaling Technology (Beverly, MA, USA). These were detected after binding to goat anti-mouse IgG-HRP secondary antibodies (1:10,000). The secondary antibodies (Cat Nos. sc-516102 and sc-2004) were purchased from Santa Cruz Biotechnology (Dallas, TX, USA).

### 4.2. Preparation of a Whole-Plant Extract of Circaea mollis

The plant extract (KPM011-095) used in this study was purchased from the Natural Product Central Bank at the Korea Research Institute of Bioscience and Biotechnology (Daejeon, Republic of Korea). The plant was collected from Ulleung-gun, Gyeongsangbuk-do, Korea, in 2001, a voucher specimen (KRIB 0000753) of which is maintained in the herbarium of the Korea Research Institute of Bioscience and Biotechnology. The plant (60 g) was dried in the shade and then powdered. Following the addition of 1 L of methanol 99.9% (HPLC-grade), the preparation was extracted at room temperature through 30 cycles (ultrasonication at 40 kHz and 1500 W for 15 min per cycle, with standing for 120 min between cycles) using an ultrasonic extractor (SDN-900H; SD-Ultrasonic Co., Ltd., Seoul, Republic of Korea). The crude extract thus obtained was subsequently filtered (Qualitative Filter No.100; Hyundai Micro Co., Ltd., Seoul, Republic of Korea) and dried under reduced pressure to yield the final *C. mollis* extract (5.09 g).

### 4.3. Liquid Chromatography–Mass Spectrometry (LC-MS)

With reference to the information provided by the Natural Product Central Bank, we prepared an analytical sample of the CS&Z extract by diluting with 99.9% methanol to produce a concentration of 1000 ppm. The detailed procedure closely followed the methodology outlined by Kim et al. [38]. MassHunter Qualitative Analysis software (version B 07.00; Agilent Technologies, Santa Clara, CA, USA) was used for the analysis of the results obtained.

### 4.4. Cell Lines

Colorectal cancer cell lines (HCT116^p53+/+^, HT29, and DLD-1 cells) were purchased from the Korean Cell Line Bank (Seoul, Republic of Korea). All cell lines were cultured using RPMI-1640 medium containing 1% penicillin–streptomycin and 10% FBS, and incubated at 37 °C in a 5% CO_2_ atmosphere.

### 4.5. Cytotoxicity Assay

The potential cytotoxicity of CS&Z was assessed using a 3-(4,5-dimethylthiazol-2-yl)-2,5-diphenyltetrazolium bromide (MTT) assay. Colorectal cancer cell lines were seeded in 96-well plates (1 × 10^4^ cells/well) and treated with different concentration of the CS&Z extract for 24 h. Following the sequential addition of MTT solution and dimethyl sulfoxide (DMSO), absorbance was measured at 540 nm using a Multiskan^TM^ GO Microplate Spectrophotometer Reader (Thermo Fisher Scientific, Mississauga, ON, Canada). The specific MTT assay procedure followed that described previously [13].

### 4.6. Colony Formation Assay

To examine the proliferation of CS&Z-treated colorectal cancer cells, we performed a colony formation assay. HCT116^p53+/+^, HT29, and DLD-1 cells were seeded in six-well plates (2 × 10^5^ cells/well) and treated with selected concentrations of CS&Z for 24 h. Thereafter, the cells were reseeded in new six-well plates (1 × 10^5^ cells/well) and incubated at 37 °C under a 5% CO_2_ atmosphere for 14 days. Afterward, the cells were fixed and stained using a Diff-Quick kit (Sysmex Corporation, Kobe, Hyogo, Japan).

### 4.7. Western Blotting

To assess the effects of the CS&Z extract on colorectal cancer cells at the protein level, we performed western blotting. The cells were seeded in six-well plates for 24 h, treated with CS&Z for 24 h, and then harvested. Following lysis using a lysis buffer, protein concentrations were determined using the Bradford assay. Subsequently, lysate samples were run via SDS-PAGE, and the separated proteins were transferred to nitrocellulose membranes. Having initially blocked the membranes with 3% bovine serum albumin in TBS-T for 1 h, the membranes were incubated overnight with the respective primary antibodies. The following day, the membranes were incubated with the secondary antibodies for 1 h, and protein detection was performed. The detailed procedures have been described previously [13].

### 4.8. c-Myc Stability Assay Using Cycloheximide

HCT116^p53+/+^, HT29, and DLD-1 cells were seeded in six-well plates (2 × 10^5^ cells/well) and treated with CS&Z for 24 h. Following treatment, the cells were treated with 50 µg/mL cycloheximide for different times (0, 30, 60, and 90 min) and then harvested to assess c-Myc stability with the protein level of c-Myc via western blotting.

### 4.9. Serum Stimulation

Colorectal cancer cell lines (HCT116^p53+/+^, HT29, and DLD-1 cells) were seeded in six-well plates (2 × 10^5^ cells/well). We treated serum-free medium containing 0.2% FBS for 24 h to starve the cells. Thereafter, the cells were treated with CS&Z diluted in serum-free medium containing 20% FBS and harvested at 0, 6, 12, and 24 h, with the respective levels of c-Myc, and β-actin protein being determined by western blotting.

### 4.10. Terminal Deoxynucleotidyl Transferase Nick-End-Labeling Assay

To establish whether CS&Z induces apoptosis in colorectal cancer cell lines, we performed a terminal deoxynucleotidyl transferase nick-end-labeling (TUNEL) assay using a DeadEnd^TM^ Fluorometric TUNEL system kit (Promega, Madison, WI, USA). The specific TUNEL assay procedure used has been described previously [13]. Briefly, cells were seeded in four-well confocal slides (5 × 10^4^ cells/well) and treated with CS&Z for 24 h. Following treatment, the cells were sequentially processed via fixing, permeabilizing, equilibrating, and labeling. Having stopped the labeling reaction, the cells were mounted and stained with DAPI. The stained cells were analyzed using a CELENA^TM^ S Digital Imaging System (Logos Biosystems, Inc., Anyang-si, Gyeonggi-do, Republic of Korea).

### 4.11. siRNA Transfection

Colorectal cancer cell lines (HCT116^p53+/+^ and DLD-1 cells) were seeded in six-well plates (7 × 10^4^ cells/well) and subsequently transfected with control siRNA (Bioneer, Daejeon, Republic of Korea) or RPL5 siRNA (Bioneer, Daejeon, Republic of Korea), which had initially been mixed with INTERFERin transfection reagent (Polyplus-transfection Inc., New York, NY, USA) for 15 min. Following incubation at 37 °C for 48 h, the cells were treated with 100 µg/mL CS&Z for 24 h.

### 4.12. Statistical Analysis

Quantitative differences between groups were analyzed using an unpaired t-test (two-tailed) in the GraphPad Prism 8 software package (GraphPad Software, San Diego, CA, USA). Statistical significance was set at a *p*-value < 0.05. Data are presented as the mean values of three independent replicates.

## 5. Conclusions

CS&Z has been demonstrated to have anticancer effects by inducing apoptosis in colorectal cancer cells. CS&Z inhibited the expression of proteins in the order of pro-Caspase 3, pro-PARP, and cleaved-PARP in cells. Furthermore, apoptosis induction was confirmed through the TUNEL assay. Building upon previous research that indicated the involvement of MID1IP1 in CNOT2 inhibition and apoptosis induction, we observed the suppression of protein levels of these oncogenes. The potential to induce G1/S phase arrest in the cell cycle was suggested by inhibiting the protein levels of cell cycle-related factors, including Cyclin E1, Cyclin D1, and CDK6. The most noteworthy finding is that CS&Z mediates the inhibition of c-Myc activity through RPL5 and downregulates c-Myc expression when stimulated with 20% FBS and in the presence of CHX. Furthermore, when CS&Z was combined with DOX and 5-FU, the protein levels of apoptotic factors and oncogenes exhibited a more significant increase compared with individual treatment with these existing therapies. While this study offers the possibility of CS&Z becoming a new candidate for colorectal cancer treatment, it should be noted that the research was conducted in vitro. Further research is necessary to validate these findings.

## Figures and Tables

**Figure 1 ijms-26-04664-f001:**
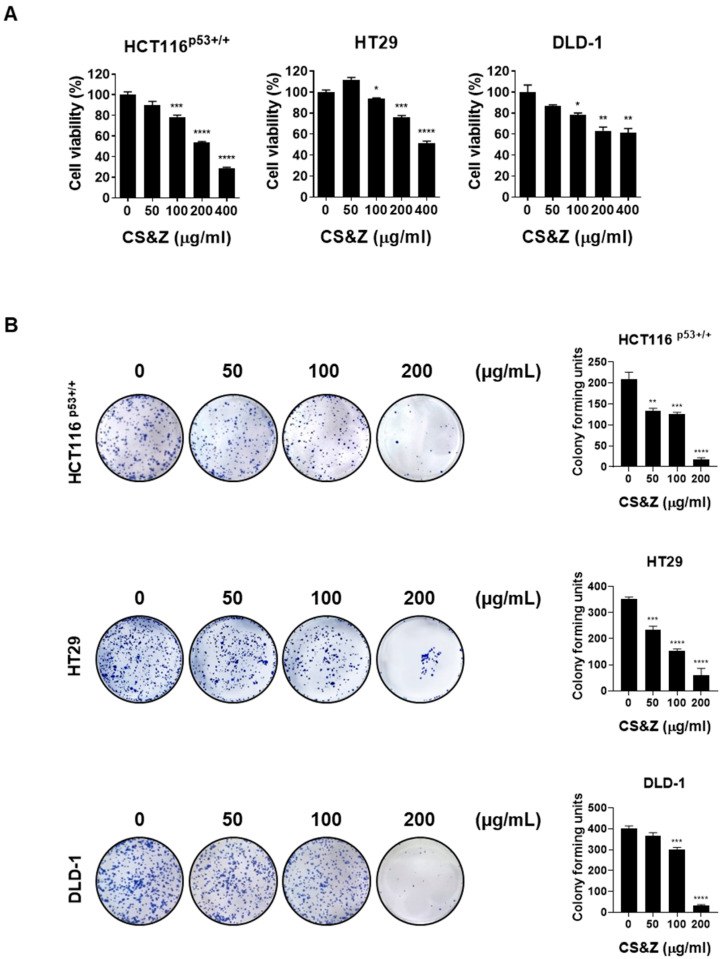
Effects of CS&Z on the viability and proliferation of colorectal cancer cells. (**A**) Colorectal cancer cell lines (HCT116^p53+/+^, HT29, and DLD-1) were treated with CS&Z for 24 h and cell viability was confirmed by MTT assay. The results revealed that cell viability was significantly inhibited in a dose-dependent manner (0, 50, 100, 200, and 400 μg/mL). (**B**) Colorectal cancer cells were treated with CS&Z for 24 h, and the effects on proliferation were assessed via colony formation assay. All data are presented as the mean ± SEM. n = 3–4. * *p* < 0.05, ** *p* < 0.005 and *** *p* < 0.001, and **** *p* < 0.0001 with the control group.

**Figure 2 ijms-26-04664-f002:**
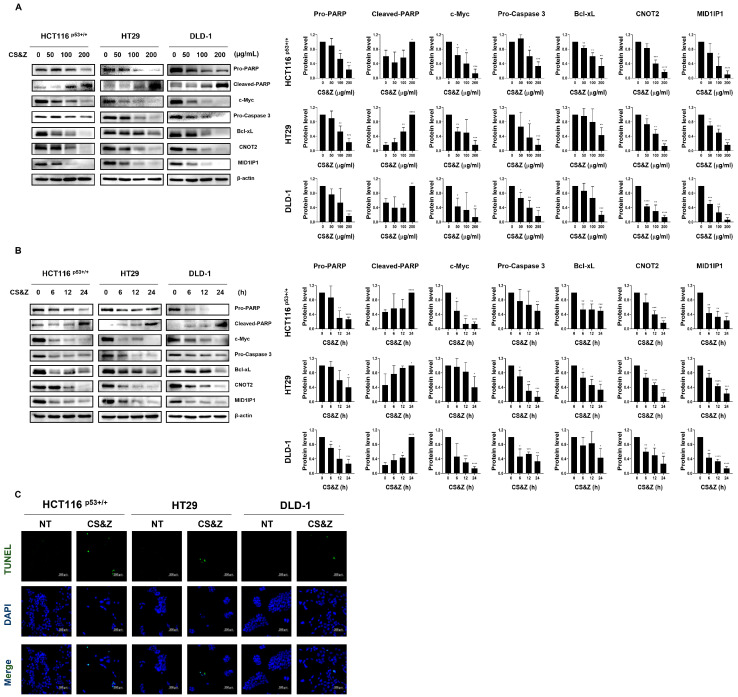
Effects of CS&Z on oncogenes and apoptotic factors in colorectal cancer cells. (**A**) Colorectal cancer cells were treated with CS&Z at various concentrations (0, 50, 100, and 200 μg/mL) for 24 h, and (**B**) treated with CS&Z at 200 μg/mL for different time intervals (0, 6, 12, and 24 h. We quantified the relative expression of each factor compared with that of the β-actin protein using Image J (version 1.44p). (**C**) We confirmed the occurrence of apoptosis in colorectal cancer cells treated with 200 μg/mL CS&Z using a TUNEL assay. For all images, the magnification is 200×. Scale bar, 200 μm. All data are presented as the mean ± SEM. n = 3. * *p* < 0.05, ** *p* < 0.005, *** *p* < 0.001, and **** *p* < 0.0001 with the control group.

**Figure 3 ijms-26-04664-f003:**
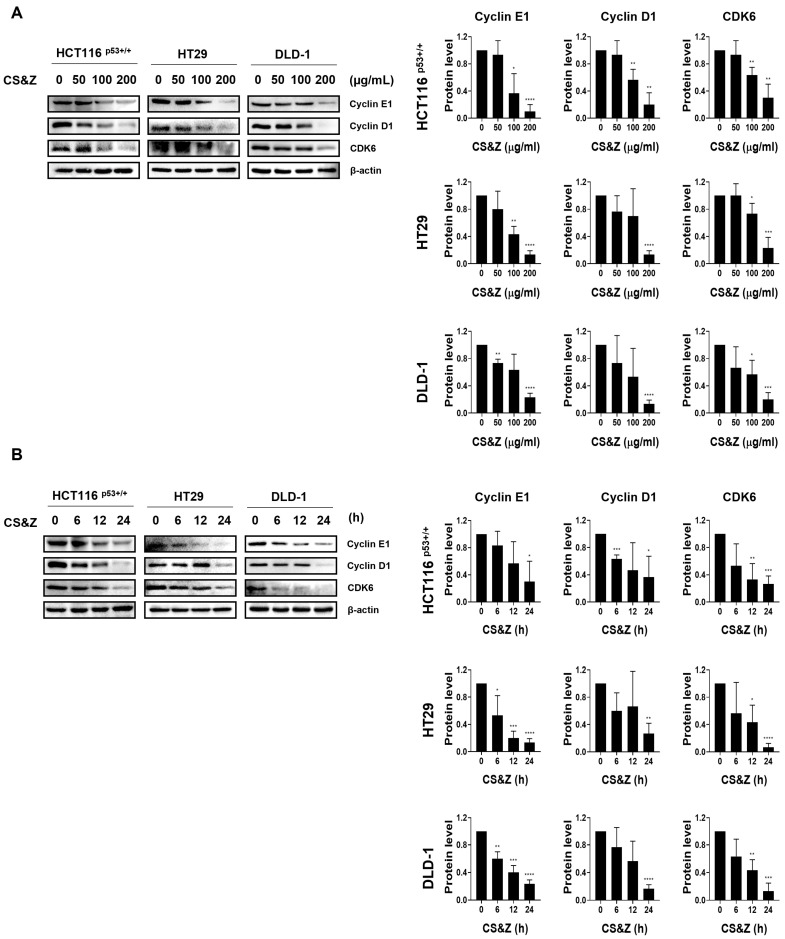
Effects of CS&Z on cell cycle-related factors in colorectal cancer cells. (**A**) We treated HCT116^p53+/+^, HT29, and DLD-1 cells with different concentrations of CS&Z (0, 50, 100, and 200 μg/mL) for 24 h and thereafter determined the expression of G1/S phase-related factors. (**B**) We treated HCT116^p53+/+^, HT29, and DLD-1 cells with 200 μg/mL CS&Z for different time intervals (0, 6, 12, and 24 h) and examined the expression of G1/S phase-related factors. All data are presented as the mean ± SEM. n = 3. * *p* < 0.05, ** *p* < 0.005, *** *p* < 0.001, and **** *p* < 0.0001 with the control group.

**Figure 4 ijms-26-04664-f004:**
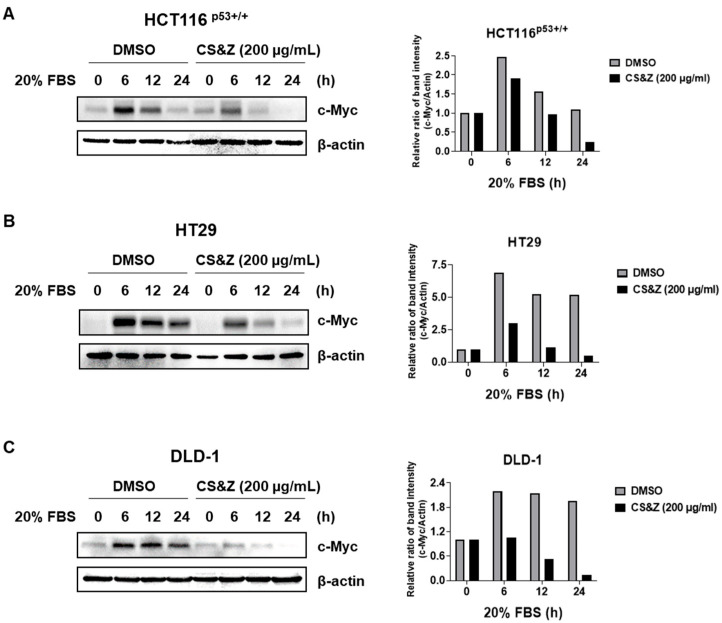
Effects of CS&Z on the sensitivity of c-Myc to serum stimulation in colorectal cancer cells. We treated starved (**A**) HCT116^p53+/+^, (**B**) HT29, and (**C**) DLD-1 cells induced by 0.2% FBS with DMSO or 200 μg/mL CS&Z diluted in 20% FBS for different time intervals (0, 6, 12, and 24 h). Following treatment, we confirmed that CS&Z regulates the expression of c-Myc in serum-stimulated colorectal cancer cells.

**Figure 5 ijms-26-04664-f005:**
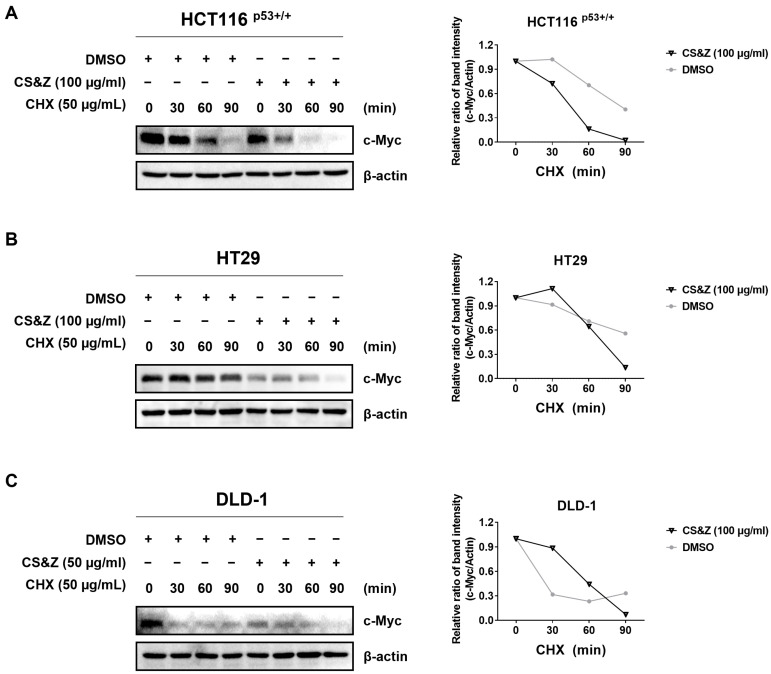
Effects of CS&Z on c-Myc stability in colorectal cancer cells. (**A**) HCT116^p53+/+^ cells and (**B**) HT29 cells were treated with DMSO or 100 μg/mL CS&Z for 24 h and subsequently exposed to cycloheximide (CHX) for different time intervals (0, 30, 60, and 90 min). (**C**) The same procedure was conducted with DLD-1 cells, although these cells were treated with 50 μg/mL CS&Z. Furthermore, in each treatment group, c-Myc expression was quantified using Image J (version 1.44p), normalized to the results at 0 min.

**Figure 6 ijms-26-04664-f006:**
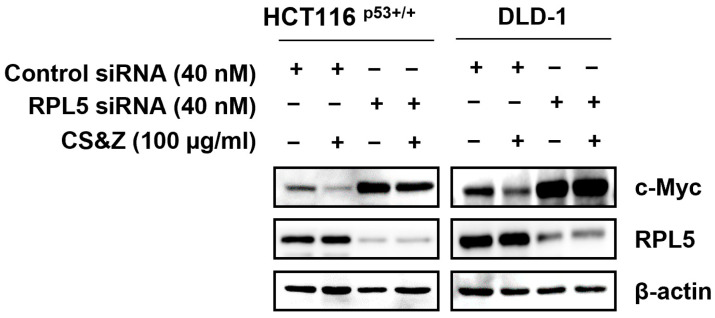
Effects of RPL5 knockdown on c-Myc inhibition induced by CS&Z in colorectal cancer cells. HCT116^p53+/+^ and DLD-1 cells were exposed to control siRNA or RPL5 siRNA for 48 h, followed by treatment with 100 μg/mL CS&Z for 24 h.

**Figure 7 ijms-26-04664-f007:**
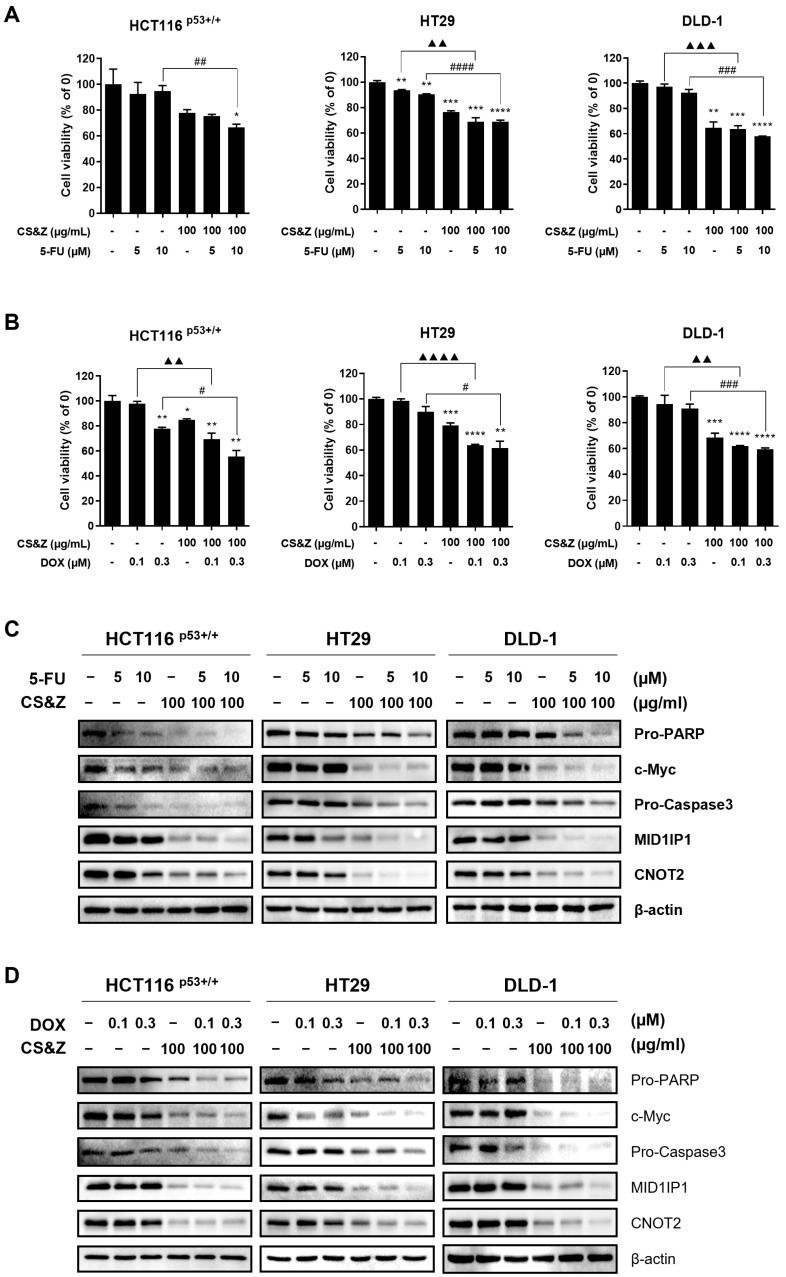
Effects of co-treatment with CS&Z and 5-FU or DOX on colorectal cancer cells. (**A**,**B**) We examined the viability of colorectal cancer cells using an MTT assay after co-treating cells with CS&Z (100 μg/mL) and 5-FU (0, 5 and 10 μM) or DOX (0, 0.1 and 0.3 μM) for 24 h. The results revealed a synergistic reduction in the viability of cells co-administered CS&Z and either 5-FU or DOX. (**C**,**D**) All data are presented as the mean ± SEM. n = 3. * *p* < 0.05, ** *p* < 0.005 and *** *p* < 0.001, **** *p* < 0.0001 compared with the no-treatment concentration group. ^▲▲^ *p* < 0.005, ^▲▲▲^ *p* < 0.001, and ^▲▲▲▲^ *p* < 0.0001 compared with only 5-FU (5 μM) or DOX (0.1 μM) group. # *p* < 0.05, ## *p* < 0.005 ### *p* < 0.001, and #### *p* < 0.0001 compared with only 5-FU (10 μM) or DOX (0.3 μM) group.

**Figure 8 ijms-26-04664-f008:**
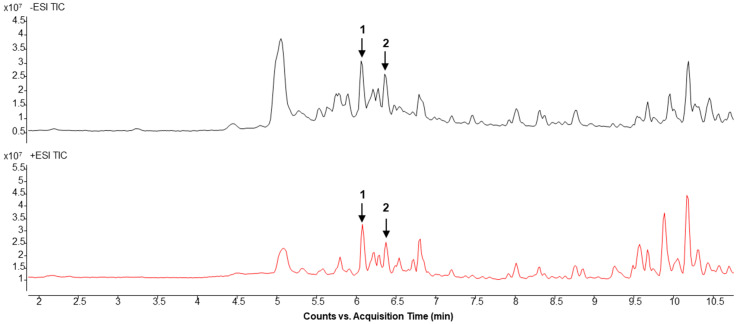
LC-MS chromatogram of a methanolic extract of CS&Z. The chromatogram of CS&Z is displayed, and each peak, corresponding to the detected active components, is labeled with a number. The active constituents corresponding to the two peaks are as follows. 1: galuteolin, isoorientin, orientin, and astragalin; 2: Vitexin.

**Table 1 ijms-26-04664-t001:** List of the active components detected in a methanolic extract of CS&Z based on liquid chromatography–mass spectrometry.

No.	Compound	R.T (min)	Mass	Molecular Formula	Experimental Mass (*m*/*z*)	Selected Ion Species
1	Galuteolin Isoorientin Orientin Astragalin	6.062	448.1006	C_21_H_20_O_11_	(+) 449.0662 (−) 447.0973	(M + H)+ (M − H)−
2	Vitexin	6.360	432.1056	C_21_H_20_O_10_	(+) 433.1038 (−) 431.1003	(M + H)+ (M − H)−

## Data Availability

Data are contained within the article.

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
