# Peer review of "Circaea mollis Siebold & Zucc. Induces Apoptosis in Colorectal Cancer Cells by Inhibiting c-Myc Through the Mediation of RPL5"

_ijms, 2025, doi:10.3390/ijms26104664_

Round 1

Reviewer 1 Report

Comments and Suggestions for Authors

Dear Authors,

First of all, congratulations for your interesting work. I hope that my hints will help you in the next steps of improvement and the final manuscript will be really valuable for the readers.

There are several grammar and punctation mistakes (such as double space, double dot or no at all) and some typos - even if they do not change the value of the manuscript, I'd like to urge you to correct these imperfections.

Moreover, gene names should be written in italics, according to the rules of genetic guidelines, Please, familiarise yourself with the rules and change the manuscript accordingly. Examples of rules summary can be found on websites such as: https://www.gmb.org.br/geneprotein-nomenclature-guidelines

or https://academic.oup.com/molehr/pages/Gene_And_Protein_Nomenclature

However, the most important issue is the name of the plant - until the line74 we had no idea what is this Circaea mollis Ciebold and Zucc. Especially using it in the title of the paper may not be the best idea - why cannot you simplify the name to C.mollis instead of using abbreviation CS&Z? It makes reading more complicated and understading of the paper too. Please, consider making it easier, including the title. Also, it should be mentioned that it is a plant, both in the title and the abstract (line 18 maybe?). 

It might be a good idea to explain and develop the information in the lines 39-41, in its current version it is incomprehensible. 

In line 56 there is an information that this study is focused on c-myc - why exactly this oncogene? There are many other oncogenes and it is not clear why exactly this one has been selected. 

Line 84 mentioned Hani - what is it? 

Lines 96-98 should not be there, I pressume. 

Not every abbreviation used has been explained, correct please and additionally please add them to the abbreviation list at the end of the manuscript. Especially when it comes to the results and materials and methods sections. 

Figures 2 and 3 definitely should be bigger, in their current form it is difficult to understand nor read them. 

There is no information about the component of the plant that may posses the strongest anticancer impact - it is very difficult to extract it and it's good you have mentioned this in lines 278 - 280, can you extract such information and add a paragraph about limitations of the study? Similarly information from lines 438-9 about the in vitro type of reseach - this should be highlighted much earlier, ideally in abstract too. 

Finally, I would like to encourage you to come back to the abstract part of your manuscript. In a current form it looks like a conference note or an abstract of a poster, it is not encouraging readers to dig deeper into your research, it seems not interesting. An abstract should have more popular-science style, to show the importance and significance of your work, not solely a mini-summary of what you have done. I strongly encourage you to rewrite this section.

Author Response

Reviewer #1:

Dear Reviewer,

We sincerely appreciate your time and effort in reviewing our manuscript. Your valuable insights and constructive feedback have significantly contributed to improving the quality of our work. We have carefully addressed all the comments and revised the manuscript accordingly.

Thank you for your thoughtful review and consideration.

Best regards,

Hyeung-Jin Jang

Comment 1: There are several grammar and punctation mistakes (such as double space, double dot or no at all) and some typos - even if they do not change the value of the manuscript, I'd like to urge you to correct these imperfections.

Response 1: We have carefully reviewed and corrected all grammar, punctuation, and typographical errors throughout the manuscript. These changes are reflected in the revised version.

Comment 2: Moreover, gene names should be written in italics, according to the rules of genetic guidelines, Please, familiarise yourself with the rules and change the manuscript accordingly. Examples of rules summary can be found on websites.

Response 2: Thank you for your helpful comment. We are aware of the general convention of italicizing gene names while keeping protein names in regular font. In preparing our manuscript, we referred to a variety of peer-reviewed articles including (1), where gene names (e.g., c-Myc, CNOT2, MID1IP1) are presented in standard font. For consistency with this style, we have retained the regular font formatting in our manuscript.

Nevertheless, we have carefully reviewed the usage of gene names in our manuscript and have applied italic formatting where appropriate. For example, the reference to MYC in Line 267 has been corrected to italics, as it pertains to the gene, not the protein. If we have misunderstood or overlooked any part of the guideline, we would sincerely appreciate further clarification and will gladly make the necessary corrections.

Comment 3: However, the most important issue is the name of the plant - until the line74 we had no idea what is this Circaea mollis Ciebold and Zucc. Especially using it in the title of the paper may not be the best idea - why cannot you simplify the name to C.mollis instead of using abbreviation CS&Z? It makes reading more complicated and understading of the paper too. Please, consider making it easier, including the title. Also, it should be mentioned that it is a plant, both in the title and the abstract (line 18 maybe?). 

Response 3: Thank you for this important suggestion. We understand the reviewer’s concern regarding the abbreviation CS&Z and the clarity of the plant identity. We chose to use the abbreviation CS&Z after the first full mention of Circaea mollis Siebold & Zucc. to ensure consistency and brevity throughout the manuscript. While we acknowledge that the abbreviation may be unconventional, we maintained it to avoid redundancy. We have clarified in the Introduction that this is a plant extract and ensured that the species name is defined (lines 18, 75).

Comment 4: It might be a good idea to explain and develop the information in the lines 39-41, in its current version it is incomprehensible. 

Response 4: Thank you for pointing this out. We revised the sentence to improve clarity and better convey the epidemiological trend.

Comment 5: In line 56 there is an information that this study is focused on c-Myc – why exactly this oncogene? There are many other oncogenes and it is not clear why exactly this one has been selected.

Response 5: Thank you for this important observation. In the revised manuscript (Lines 53–57), we have expanded our explanation to clarify the rationale behind focusing on c-Myc. This oncogene was selected due to its critical role in regulating cell proliferation, apoptosis, and ribosome biogenesis, as well as its high frequency of overexpression in colorectal cancer. Moreover, its direct regulatory relationship with ribosomal protein L5 (RPL5), a key target of this study, made c-Myc a mechanistically relevant and biologically meaningful focus for evaluating the effects of CS&Z. We believe this revision provides a clearer justification for our choice.

Comment 6: Line 84 mentioned Hani - what is it? 

Response 6: Thank you for your comment. “Hani” refers to an ethnic minority group in China. While we did not include an explicit explanation in the manuscript, we followed common usage in ethnopharmacology literature, where the term is often used without further definition. For example, the study by (2) uses the term “Hani medicine” without elaboration, as it is familiar within the context of traditional Chinese ethnomedicine.

Comment 7: Lines 96-98 should not be there, I pressume. 

Response 7: Thank you for pointing this out. We agree that the content in Lines 96–98 was not relevant to the main text and may have been inadvertently included. We have removed these lines in the revised manuscript.

Comment 8: Not every abbreviation used has been explained, correct please and additionally please add them to the abbreviation list at the end of the manuscript. Especially when it comes to the results and materials and methods sections. 

Response 8: Thank you for your careful review. We have thoroughly checked the manuscript to ensure that all abbreviations are clearly defined upon their first appearance in the text, particularly in the Results and Materials and Methods sections. In addition, we updated the abbreviation list at the end of the manuscript to include all relevant terms.

Comment 9: Figures 2 and 3 definitely should be bigger, in their current form it is difficult to understand nor read them. 

Response 9: Thank you for the valuable feedback. We have increased the size and resolution of Figures 2 and 3 in the revised manuscript to improve clarity and readability. We believe the updated figures now allow for easier interpretation of the data.

Comment 10: There is no information about the component of the plant that may posses the strongest anticancer impact - it is very difficult to extract it and it's good you have mentioned this in lines 278 - 280, can you extract such information and add a paragraph about limitations of the study? Similarly information from lines 438-9 about the in vitro type of reseach - this should be highlighted much earlier, ideally in abstract too. 

Response 10: We thank the reviewer for this thoughtful comment. In the revised manuscript, we have acknowledged that although five compounds were identified in peak 1 of the LC-MS analysis, their individual anticancer activities remain unconfirmed. We have clarified this as a key limitation of our study and added a statement indicating that future research will focus on fractionation and isolation of active constituents to identify the compounds responsible for the observed effects (Lines 261–265). Additionally, as suggested, we revised the abstract and discussion to clearly indicate that the current study was conducted entirely in vitro. This change ensures transparency about the experimental context and helps readers properly interpret the scope and applicability of our findings. 

Comment 11: I would like to encourage you to come back to the abstract part of your manuscript. In a current form it looks like a conference note or an abstract of a poster, it is not encouraging readers to dig deeper into your research, it seems not interesting. An abstract should have more popular-science style, to show the importance and significance of your work, not solely a mini-summary of what you have done. I strongly encourage you to rewrite this section.

Response 11: Thank you for the helpful suggestion. We revised the abstract to better emphasize the significance of our findings, including the therapeutic relevance of CS&Z, its unique mechanism via the RPL5–c-Myc axis, and its potential synergy with standard treatments. We believe the new version more effectively conveys the impact of our work.

References

  1. Jung JH, Lee HJ, Kim JH, Sim DY, Im E, Kim S, et al. Colocalization of MID1IP1 and c-Myc is Critically Involved in Liver Cancer Growth via Regulation of Ribosomal Protein L5 and L11 and CNOT2. Cells. 2020;9(4).
  2. Park JH, Son YJ, Lee CH, Nho CW, Yoo G. Circaea mollis Siebold & Zucc. Alleviates postmenopausal osteoporosis in a mouse model via the BMP-2/4/Runx2 pathway. BMC Complementary Medicine and Therapies. 2020;20(1):123.

Reviewer 2 Report

Comments and Suggestions for Authors

The manuscript investigates the anticancer effects of Circaea mollis on colorectal cancer cells, demonstrating that CS&Z induces apoptosis and G1/S cell cycle arrest through RPL5-mediated suppression of c-Myc. The study is well-structured, presents novel mechanistic insights, and shows that CS&Z enhances the efficacy of chemotherapeutics like 5-FU and doxorubicin. Minor revisions are needed to improve clarity, statistical rigor, methodological detail, and contextual framing of the findings. Overall, this work provides significant findings that support the potential of CS&Z as a complementary therapy for colorectal cancer. However, while the overall scientific quality is commendable, several critical issues including insufficient mechanistic resolution of active compounds, lack of direct cell cycle quantification, statistical shortcomings, and the need for clearer translational framing, must be addressed before acceptance.

Here are my detailed comments and suggestions:

  1. While the study demonstrates novelty in targeting c-Myc via RPL5, more explicit comparison with similar natural products or previous studies on CS&Z would help to contextualize the novelty and highlight its translational potential.

  1. The study is entirely in vitro. While this is acceptable for preliminary discovery, there is insufficient discussion of in vivo relevance or potential pharmacokinetic/therapeutic limitations of CS&Z in physiological settings. Include a discussion of in vivo feasibility, bioavailability, and potential future animal studies.

  1. Additional details on extract preparation (e.g., batch reproducibility, % yield consistency) would aid reproducibility.

  1. Flow cytometry is mentioned as a needed follow-up; it would have strengthened claims regarding cell cycle arrest if included.

  1. The five flavonoids identified by LC-MS are not individually tested or validated as contributors to the observed activity. The authors should clarify whether these constituents were evaluated for activity or if further work is planned to isolate active compounds.

  1. G1/S arrest is inferred from Western blot analysis without direct cell cycle quantification. nclude flow cytometry analysis or acknowledge this limitation more explicitly.

  1. Unpaired t-tests are used for multiple group comparisons instead of more robust methods like one-way ANOVA with post hoc testing. Consider reanalyzing data using appropriate statistical methods.

  1. Some methodological details are incomplete like information on the antibodies, siRNA sequences, etc.

  1. It is not clear how many biological replicates were used per Western blot. Specify whether quantification was from independent experiments or single blot replicates.

  1. Figure legends are excessively detailed and sometimes repetitive. Simplify legends by moving methodological detail to the main text.

  1. Quantification graphs (e.g., for blots) are sometimes too small or lack clarity. Improve graph clarity and consistency in axis labeling and statistical annotation.

  1. The claim that CS&Z may be used therapeutically is overstated given the lack of in vivo data. Temper conclusions and emphasize that these findings are preliminary and require further validation in animal models.

  1. Some sections (particularly the Results and Discussion) contain repetitive phrasing and could benefit from minor language polishing for conciseness and readability.

  1. A few key citations related to flavonoid pharmacokinetics and clinical translation are missing. Include discussion and references addressing the pharmacological challenges of flavonoid-based therapies (e.g., solubility, metabolism).

  1. The concluding statement could be strengthened by summarizing how CS&Z stands out among other plant-derived anticancer agents.

Author Response

Reviewer #2:

Dear Reviewer,

We sincerely appreciate your time and effort in reviewing our manuscript. Your valuable insights and constructive feedback have significantly contributed to improving the quality of our work. We have carefully addressed all the comments and revised the manuscript accordingly.

Thank you for your thoughtful review and consideration.

Best regards,

Hyeung-Jin Jang

Comment 1: While the study demonstrates novelty in targeting c-Myc via RPL5, more explicit comparison with similar natural products or previous studies on CS&Z would help to contextualize the novelty and highlight its translational potential.

Response 1: We thank the reviewer for this valuable comment. As studies on CS&Z are still limited, we focused on emphasizing its mechanism, which remains rare among plant-derived compounds. This distinction has been noted in the Discussion (Lines 302–305) to highlight its translational potential.

Comment 2: The study is entirely in vitro. While this is acceptable for preliminary discovery, there is insufficient discussion of in vivo relevance or potential pharmacokinetic/therapeutic limitations of CS&Z in physiological settings. Include a discussion of in vivo feasibility, bioavailability, and potential future animal studies.

Response 2: We thank the reviewer for this critical comment. In the revised manuscript, we added a new paragraph to the Discussion section (Lines 306–308) addressing the limitations of in vitro studies and emphasizing the need for future in vivo validation. These revisions provide a clearer framework for interpreting our findings and outline a logical next step for future research.

Comment 3: Additional details on extract preparation (e.g., batch reproducibility, % yield consistency) would aid reproducibility.

Response 3: We appreciate the reviewer’s comment regarding extract preparation. In this study, the CS&Z extract was obtained from the Korea Plant Extract Bank, which provides standardized plant extracts prepared under controlled and reproducible conditions. As such, detailed extraction protocols and quality control are managed by the institution, ensuring consistency and batch reproducibility.

Comment 4: Flow cytometry is mentioned as a needed follow-up; it would have strengthened claims regarding cell cycle arrest if included.

Response 4: Thank you for this valuable suggestion. We agree that the inclusion of flow cytometry data would have further strengthened the conclusions. Although it was not feasible within the scope of the current study, we have highlighted the potential value of such experiments as part of future investigations.

Comment 5: The five flavonoids identified by LC-MS are not individually tested or validated as contributors to the observed activity. The authors should clarify whether these constituents were evaluated for activity or if further work is planned to isolate active compounds.

Response 5: We thank the reviewer for this important comment. In the revised manuscript (Lines 261–265), we have clarified that although five compounds were detected in peak 1 of the LC-MS analysis, their individual anticancer activities have not yet been evaluated. We have acknowledged this as a limitation of the study and added a statement indicating that future fractionation and isolation studies are planned to identify the specific active compounds responsible for the observed effects.

Comment 6: G1/S arrest is inferred from Western blot analysis without direct cell cycle quantification. nclude flow cytometry analysis or acknowledge this limitation more explicitly.

Response 6: We appreciate the reviewer’s insightful comment. In the revised manuscript, we clarified that our conclusion regarding G1/S phase arrest is based on indirect evidence obtained from Western blot analysis of cell cycle–related proteins. We also explicitly acknowledged this as a limitation and emphasized the need for flow cytometry in future studies to confirm the precise point of cell cycle arrest. These revisions have been made in Lines 278–283.

Comment 7: Unpaired t-tests are used for multiple group comparisons instead of more robust methods like one-way ANOVA with post hoc testing. Consider reanalyzing data using appropriate statistical methods.

Response 7: We thank the reviewer for this insightful comment. In this study, most of the comparisons were designed to evaluate differences between two experimental conditions, and therefore unpaired t-tests were applied. To maintain consistency across figures and ensure interpretability, we applied the same statistical method throughout the analyses.

Comment 8: Some methodological details are incomplete like information on the antibodies, siRNA sequences, etc.

Response 8: We thank the reviewer for this comment. The relevant information, including the sources and catalog numbers of antibodies as well as the sources of siRNA, was already provided in the Materials and Methods sections of the original manuscript.

Comment 9: It is not clear how many biological replicates were used per Western blot. Specify whether quantification was from independent experiments or single blot replicates.

Response 9: We thank the reviewer for this comment. The manuscript already states that “data are presented as the mean values of three independent replicates,” which applies to all quantified Western blot results. We hope this clarification resolves any confusion.

Comment 10: Figure legends are excessively detailed and sometimes repetitive. Simplify legends by moving methodological detail to the main text.

Response 10: Thank you for this helpful suggestion. Upon review, we found that most figure legends were consistent with standard practice for clarity and completeness. However, we agree that some methodological details were better suited to the main text, and we have accordingly revised select figure legends to improve conciseness and avoid redundancy.

Comment 11: Quantification graphs (e.g., for blots) are sometimes too small or lack clarity. Improve graph clarity and consistency in axis labeling and statistical annotation.

Response 11: We appreciate the reviewer’s suggestion. In response, we have enlarged the quantification graphs in Figures 2 and 3 and adjusted axis labels, font size, and statistical annotations for improved clarity and consistency. These changes enhance the visual presentation and ensure accurate data interpretation.

Comment 12: The claim that CS&Z may be used therapeutically is overstated given the lack of in vivo data. Temper conclusions and emphasize that these findings are preliminary and require further validation in animal models.

Response 12: We thank the reviewer for this important comment. In the revised manuscript, we have adjusted the tone of our conclusions to more accurately reflect the preliminary nature of the findings. Specifically, we included a statement at the end of the Discussion (Lines 306–308) noting that the current study was conducted in vitro and that further in vivo validation will be necessary to assess the therapeutic applicability of CS&Z. We believe this revision sufficiently addresses the reviewer’s concern regarding the scope and strength of our conclusions.

Comment 13: Some sections (particularly the Results and Discussion) contain repetitive phrasing and could benefit from minor language polishing for conciseness and readability.

Response 13: Thank you for this helpful suggestion. We reviewed the Results and Discussion sections and revised several instances of repetitive phrasing to improve clarity and readability. These edits included reducing redundancy in sentence structure and improving the flow of ideas while preserving the original scientific content.

Comment 14: A few key citations related to flavonoid pharmacokinetics and clinical translation are missing. Include discussion and references addressing the pharmacological challenges of flavonoid-based therapies (e.g., solubility, metabolism).

Response 14: We thank the reviewer for this important suggestion. We fully acknowledge that pharmacokinetic challenges are critical considerations for the clinical translation of flavonoid-based compounds. However, as this study focused on the in vitro identification of anticancer mechanisms, an in-depth pharmacokinetic analysis was beyond the scope of the current work. Follow-up studies are currently being planned to investigate the bioavailability and in vivo behavior of the identified compounds, which will include detailed pharmacokinetic profiling and formulation strategies.

Comment 15: The concluding statement could be strengthened by summarizing how CS&Z stands out among other plant-derived anticancer agents.

Response 15: Thank you for the insightful comment. We added a concluding sentence in the Discussion section emphasizing how CS&Z differs from other plant-derived anticancer agents, particularly through its targeting of the RPL5–c-Myc axis and potential synergy with standard chemotherapeutics.
